# Decision Mamba: Reinforcement Learning via Hybrid Selective Sequence Modeling

**Sili Huang**[1] **Jifeng Hu**[2] **Zhejian Yang**[3] **Liwei Yang**[4] **Tao Luo**[5]
**Hechang Chen**[6*] **Lichao Sun**[7] **Bo Yang**[8*]

[1,8]Key Laboratory of Symbolic Computation and Knowledge Engineering of Ministry of Education
[1,2,3,6]School of Artificial Intelligence, Jilin University, China
[4,5] Institute of High Performance Computing, Agency for Science, Technology and Research, Singapore
[7]Lehigh University, Bethlehem, Pennsylvania, USA

## Abstract

Recent works have shown the remarkable superiority of transformer models in reinforcement learning (RL), where the decision-making problem is formulated as sequential generation. Transformer-based agents could emerge with self-improvement in online environments by providing task contexts, such as multiple trajectories, called in-context RL. However, due to the quadratic computation complexity of attention in transformers, current in-context RL methods suffer from huge computational costs as the task horizon increases. In contrast, the Mamba model is renowned for its efficient ability to process long-term dependencies, which provides an opportunity for in-context RL to solve tasks that require long-term memory. To this end, we first implement Decision Mamba (DM) by replacing the backbone of Decision Transformer (DT). Then, we propose a Decision Mamba-Hybrid (DM-H) with the merits of transformers and Mamba in high-quality prediction and long-term memory. Specifically, DM-H first generates high-value sub-goals from long-term memory through the Mamba model. Then, we use sub-goals to prompt the transformer, establishing high-quality predictions. Experimental results demonstrate that DM-H achieves state-of-the-art in long and short-term tasks, such as D4RL, Grid World, and Tmaze benchmarks. Regarding efficiency, the online testing of DM-H in the long-term task is $28\times$ times faster than the transformer-based baselines.

## 1 Introduction

Large transformer models [43] have achieved notable successes across a variety of domains, including text [4], image [9], and audio [1]. In the field of reinforcement learning (RL), large transformer models can treat RL tasks as a type of sequential prediction problem and have shown impressive results with offline training [31, 39]. However, these methods lack self-improvement when used in online environments, where online environments could differ from offline training. To overcome this, in-context RL has been proposed, which enables continued policy improvement [29].

Recent works demonstrated that in-context RL methods can automatically improve online performance by providing prompt conditions called across-episodic contexts [30]. The construction of the across-episodic context is flexible and easy to implement, such as multiple historical trajectories arranged in ascending order of returns [20]. Although no gradient updates are required for self-improvement, current in-context RL still suffers from high computational costs on long-term tasks [29]. This arises from (1) the quadratic complexity of the self-attention mechanism and (2) the multiplicative growth of the long-term sequence caused by across-episodic contexts.

---

*Corresponding Author. Bo Yang and Hechang Chen.

38th Conference on Neural Information Processing Systems (NeurIPS 2024).

Facing the challenge of handling long-term sequences, a novel foundation model called Mamba has attracted widespread attention for its ability to capture long-term dependencies with linear computational costs [17, 6]. Mamba is a state space model-based framework with good potential in natural language processing and vision tasks [47]. Motivated by the success of Mamba in language and vision modeling, it is appealing that we can also transfer this success to RL tasks. Therefore, a natural question arises:

*"Can Mamba boost in-context RL with both effectiveness and efficiency on long-term tasks?"*

Taking the famous in-context RL method Decision Transformer (DT) [29] as an example, we replace its transformer backbone with a Mamba backbone. Regarding efficiency, there is no doubt that Mamba is superior to transformers as the task length increases because Mamba uses a data-dependent selection mechanism that computes independently on each input of sequences [17]. Compared with the attention mechanism that computes input pairs, the data-dependent selection mechanism makes it easier for Mamba to handle long sequences but also introduces additional independence assumptions. In RL tasks, since there is a sequential relationship between states rather than independence, the attention mechanism may be intuitively more suitable for capturing the information between states, thereby outperforming Mamba in effectiveness.

To this end, instead of implement a Decision Mamba (DM) that simply replaces the backbone of DT, we propose the Decision Mamba-Hybrid (DM-H) with the merits of transformers and Mamba in high-quality prediction and long-term memory. Specifically, the Mamba model first generates sub-goals, represented by a vector, based on long-term contexts. Then, we combine the sub-goal and short-term contexts as a prompt condition to the transformer. The setting of sub-goals enables DM-H to leverage Mamba's ability to efficiently recall long-term contexts while using the transformer to predict high-quality actions. However, it remains to be seen whether the transformer will benefit from sub-goals or ignore them and predict actions focused on short-term contexts. Therefore, we select high-value states from offline trajectories to form extra sub-goals and serve them as input conditions for the transformer. Then, the predicted actions will associate with these selected sub-goals and, in turn, encourage Mamba to generate high-value sub-goals. Our contributions are as follows:

- We investigate Mamba model compared to the transformer model in traditional RL tasks, D4RL, and find that Mamba model is more efficient but slightly inferior to the transformer model in terms of effectiveness.
- We propose DM-H, an in-context RL method that connects Mamba and the transformer with high-value sub-goals. DM-H inherits the merits of Mamba and transformers, achieving both high effectiveness and efficiency in long-term tasks.
- Our extensive experiments across Grid World, D4RL, and Tmaze reveal the superiority of DM-H over other baselines. In the online testing of long-term tasks, DM-H can be $28\times$ times faster than baselines and more than double the effectiveness.

## 2  Related Work

**Mamba for Long Sequence Modeling.** The structured State-Space Sequence (S4) model is a novel alternative to CNNs or transformers to model the long-term dependency [18]. The promising property of linearly scaling in sequence length attracts further exploration. Based on the S4, Smith et al. [40] propose a new S5 layer by introducing MIMO SSM and an efficient parallel scan into the S4 layer. Fu et al. [12] design a new SSM layer, H3, that nearly fills the performance gap between SSMs and transformers in language modeling. Mehta et al. [32] build the Gated State Space layer on S4 by introducing more gating units to improve the expressivity. Recently, Gu and Dao [17] propose a data-dependent SSM layer and build a generic language model backbone, Mamba, which outperforms transformers at various sizes on large-scale real data and enjoys linear scaling in sequence length. Later, Vision Mamba adds position coding and bidirectional scanning to extend it to visual tasks [47]. In this work, we explore transferring Mamba's success to RL, i.e., achieving high effectiveness and efficiency for long-term memory.

**Transformer for Decision-Making.** In general, reinforcement learning was proposed as a fundamental online paradigm [41]. The nature of online learning comes with some limitations when meeting the applications for which it is impossible to gather online data and learn simultaneously, such as autonomous driving. To this end, offline RL proposed that the agent can learn from a fixed offline dataset without gathering new data during learning [15, 28, 45, 27, 21, 23, 19, 24]. In the

context of offline RL, recent works explored using transformer-based policy by treating RL tasks as a type of sequential prediction problem [22]. Among them, a decision transformer [5] was proposed to model trajectories as sequences and autoregressively predict action conditioning on desired return-to-go, past states, and actions. Trajectory transformer [26] demonstrated that transformer could learn single-task policies from offline data. Subsequently, the multi-game decision transformer [31] and Gato [39] further showed that transformer-based policies could address multi-tasks in the same domain and cross-domain tasks. However, these works focused on distilling expert policies from offline data and failed to enable self-improvement like DM-H. When the offline data are sub-optimal, or the agent is required to adapt to new tasks, the multi-game decision transformers need to finetune the model parameters, while Gato is required to get prompted with expert demonstrations.

**Meta RL.** DM-H falls into the category of methods of learning to learn, which is also known as meta-learning. More precisely, recent in-context RL methods can be categorized as in-context meta-RL methods. The general idea of learning self-improvement has a long history in RL but is limited to hyper-parameters in the early stages [25]. In-context meta-RL methods [44, 10] are commonly trained in the online setting by maximizing multi-episodic value functions with memory-based architectures through environment interactions. Another online meta-RL attempts to find good network parameter initializations and then quickly adapt through additional gradient updates [11, 36]. More recently, meta-RL has seen substantial breakthroughs, from performance gains on popular benchmarks to offline settings, such as Bayesian RL [8] and optimization-based meta-RL [33]. Considering the difficulty of a completely offline setting, recent work has explored hybrid offline-online settings [46, 37]. DM-H is similar to the hybrid offline-online setting but saves more computing resources because the online phase does not involve gradient updates.

**In-Context RL.** In-context RL is the one that addresses tasks by providing prompts or demonstrations [5, 26]. By training agents at a large scale, transformer-based policies usually have the ability to learn in context [31, 39]. The learning process is performed entirely in context and does not involve parameter updates of neural networks. In this work, we consider incremental in-context RL, which involves learning from one's own behaviors in a trial-and-error manner. Laskin et al. [29] proposed Algorithm Distillation (AD), which automatically improved its performance by providing multiple historical trajectories. Subsequently, Lee et al. [30] proposed a Decision-Pretrained Transformer, which trained the agent to find optimal behaviors faster by only predicting the optimal trajectory. More recently, Hao Liu [20] further demonstrated that across-episodic contexts encourage large transformer models' emerging self-improvement behaviors. However, these methods suffer from huge computational costs as across-episodic contexts induce too-long sequences. In contrast, DM-H leverages Mamba's ability to efficiently process long-term dependencies while using the transformer to establish high-quality predictions.

## 3 Preliminaries

**Partially Observable Markov Decision Process.** We consider learning problems in the context of Partially Observable Markov Decision Processes (POMDP) represented by a tuple $\mathcal{M} = (\mathcal{S}, \mathcal{O}, \mathcal{A}, P, \mathcal{R})$. The POMDP tuple consists of states $s \in \mathcal{S}$, observations $o \in \mathcal{O}$, actions $a \in \mathcal{A}$, rewards $r \in \mathcal{R}$, and a transition probability function $P(s_{t+1}|s_t, a_t)$, where $t$ is an integer denoting the timestep. At each timestep $t$, the agent receives the observation $o_t$, selects an action $a_t \sim \pi(\cdot|o_t)$ based on its policy, and then receives the next observation $o_{t+1}$. For convenience, we uniformly use $s$ to denote the observations or states received from the environment. A trajectory is a sequence that consists of observations, actions, and rewards and is denoted by $\tau = (s_0, a_0, r_0, \ldots, s_T, a_T, r_T)$. In addition, a completion token $d_t$, a binary identifier, is used to indicate whether a trajectory ends at time $t$.

**Transformers.** The Transformer [43] architecture consists of multiple layers of self-attention operation and MLP. The self-attention begins by projecting input data $X$ with three separate matrices onto $D$-dimensional vectors called queries $Q$, keys $K$, and values $V$. These vectors are then passed through the attention function:

$$\text{Attention}(Q, K, V) = \text{softmax}(QK^T/\sqrt{D})V. \tag{1}$$

The $QK^T$ term computes an inner product between two projections of the input data $X$. The inner product is then normalized and projected back to a $D$-dimensional vector with the scaling term $V$. Transformers utilize self-attention as a core part of the architecture to process sequential data [3, 7].

Table 1: Mamba vs. Transformer on D4RL datasets.

| Environment | | HalfCheetah | | | Hopper | | | Walker2d | | |
|---|---|---|---|---|---|---|---|---|---|---|---|
| Dataset | | Med-Expert | Medium | Med-Replay | Med-Expert | Medium | Med-Replay | Med-Expert | Medium | Med-Replay |
| Effectiveness | Transformer | **94.21**± 0.46 | **42.28**± 1.18 | **41.28**± 0.21 | 108.32± 0.95 | 72.58± 0.54 | **91.32**± 0.66 | **111.36**± 0.46 | **85.96** ± 0.46 | **89.21**± 1.42 |
| | Mamba | 92.21± 0.60 | 41.92± 0.11 | 39.68± 0.13 | **110.82**± 0.56 | **73.65**± 1.23 | 82.65± 1.15 | 108.31± 0.52 | 78.26± 0.55 | 70.92± 1.21 |
| Efficiency (hour) | Transformer | | 37.10± 0.22 | | | 22.23± 0.15 | | | 33.77± 0.26 | |
| | Mamba | | **28.56**± 0.18 | | | **18.15**± 0.16 | | | **26.25**± 0.21 | |

In this work, we use GPT [38] architecture that modifies the transformer with a causal self-attention mask to focus on the previous tokens in the sequence ($j \in [1, i]$), enabling us to do autoregressive generation at test time.

**S4 and Mamba.** S4 [18] and Mamba [17] are inspired by the continuous system, which maps a $1-$D function or a sequence $x(t) \in \mathbb{R} \to y(t) \in \mathbb{R}$ through a hidden state $h(t) \in \mathbb{R}^N$. The mapping process can be represented as the following linear ordinary differential equation:

$$\begin{aligned} h'(t) &= \mathbf{A}h(t) + \mathbf{B}x(t), \\ y(t) &= \mathbf{C}h(t), \end{aligned} \quad (2)$$

where $\mathbf{A} \in \mathbb{R}^{N \times N}$ denotes the evolution parameter, $\mathbf{B} \in \mathbb{R}^{N \times 1}$ and $\mathbf{C} \in \mathbb{R}^{1 \times N}$ denote the projection parameters. For application to a discrete input sequence instead of a continuous function, S4 uses the zero-order hold to transform the continuous parameters $\mathbf{A}, \mathbf{B}$ to discrete parameters $\overline{\mathbf{A}}, \overline{\mathbf{B}}$. Then, the Equation (2) can be rewritten as:

$$\begin{aligned} h_t &= \overline{\mathbf{A}}h_{t-1} + \overline{\mathbf{B}}x_t, \\ yt &= \mathbf{C}h_t, \end{aligned} \quad (3)$$

where $\overline{\mathbf{A}} = \exp(\Delta)\mathbf{A}, \overline{\mathbf{B}} = (\delta\mathbf{A})^{-1}(\exp(\Delta)\mathbf{A} - \mathbf{I})(\Delta\mathbf{B})$, and $\Delta$ is a timescale parameter. Based on the S4 framework, Mamba introduces a data-dependent selection mechanism while leveraging a hardware-aware parallel algorithm in recurrent mode. Compared with the Transformer, the combined architecture of Mamba empowers it to capture contexts effectively and maintains computational efficiency, particularly for long sequences.

## 4 Method

In this section, we first compare Mamba and transformer models in the D4RL dataset, and investigate the potential of Mamba in RL tasks. Then, We present DM-H, which can handle long-term dependencies from contexts with high effectiveness and efficiency, as shown in Figure 1.

### 4.1 Mamba vs. Transformer in RL tasks

We first consider the Algorithm Distillation (AD) as the baseline, which is a classic in-context RL method using a transformer as the backbone [29]. AD can predict high-quality actions by recalling the historical trajectories from the context, but it also incurs higher computational costs. Under the same settings, we replace the transformer in the AD algorithm with Mamba to compare their effectiveness and efficiency.

As shown in Table 1, the simple backbone replacement did not significantly improve effectiveness. Regarding efficiency, Mamba brings predictable improvements, thus saving training time under the same settings. Compared with the attention mechanism acting on state pairs, Mamba uses a data-dependent selection mechanism acting on each state independently, which brings a more efficient method of recalling long-term memory. However, since states in RL tasks commonly exhibit sequential relationships, the attention mechanism is more suitable for capturing the information between states, thereby outperforming Mamba in terms of effectiveness. To this end, we aim at a new in-context RL approach that leverages Mamba's strengths in processing long-term memory while preserving high-quality predictions from transformers.

### 4.2 Decision Mamba-Hybrid

In-context RL can automatically improve its performance through trial-and-error when across-episodic contexts serve as prompt conditions. Specifically, an across-episodic context consisting of $n$

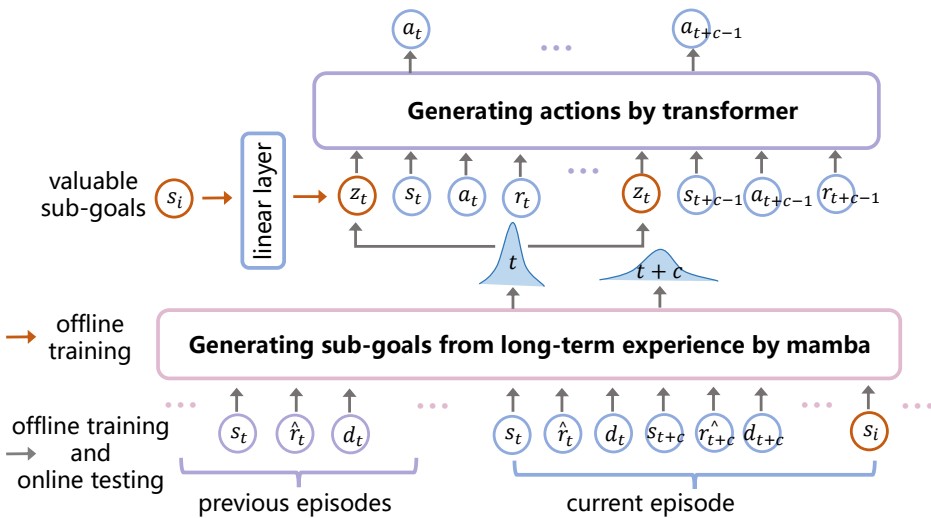

Figure 1: The architecture of DM-H. During offline training, Mamba module generates sub-goals from long-term experience, where the long-term experience consists of multiple historical trajectories arranged in ascending order of the total rewards. Based on the generated sub-goals, the transformer is required to predict better actions by supervising the expert behaviors. Meanwhile, the linear layer feeds the valuable sub-goals into the transformer module and associates them with the generated actions. During online testing, DM-H can automatically improve its performance in a trial-and-error manner without requiring gradient updates.

trajectories is represented as $(\tau^1, \tau^2, \ldots, \tau^n)$, where

$$\tau^i = (s_0^i, a_0^i, r_0^i, d_0^i, \ldots, s_T^i, a_T^i, r_T^i, d_T^i). \tag{4}$$

The trajectories are sorted according to their total rewards, i.e., $\sum_{t=0}^{T} r_t^1 \leq \sum_{t=0}^{T} r_t^2 \leq \cdots \leq \sum_{t=0}^{T} r_t^n$. With autoregressive training and generation, the transformer can uncover meaningful patterns from multiple trajectories and improve itself conditioned on experience. However, the quadratic complexity of the attention mechanism suffers from huge computational costs along with the growth in task horizon. Inspired by Mamba's success with long sequences, we propose that Mamba handles across-episodic contexts and preserves local short-term sequences for the transformer. For Mamba, we reconstruct the long-term sequences $(\tau_m^1, \tau_m^2, \ldots, \tau_m^n)$ from across-episodic contexts. Each $\tau_m^i$ is denoted as

$$\tau_m^i = (s_0^i, \hat{r}_0^i, d_0^i, s_c^i, \hat{r}_c^i, d_c^i, \ldots, s_{kc}^i, \hat{r}_{kc}^i, d_{kc}^i), \tag{5}$$

where $c$ represents the local sequence length for the transformer, $T - c \leq kc \leq T$, and $\hat{r}_c^i = \sum_{t=c}^{2c-1} r_t^i$ is the sum of $c$ steps rewards. Mamba module will generate sub-goals to prompt the transformer, where the sub-goal is represented by a vector $\mathbf{z}$ sampled from a multivariate Gaussian distribution. Then, the local short-term sequence is represented as:

$$\tau_{\mathbf{z}}^{i,j} = (\mathbf{z}_j^i, s_j^i, a_j^i, r_j^i, \mathbf{z}_j^i, s_{j+1}^i, a_{j+1}^i, r_{j+1}^i, \ldots, \mathbf{z}_j^i, s_{j+c-1}^i, a_{j+c-1}^i, r_{j+c-1}^i), \tag{6}$$

where $\tau_{\mathbf{z}}^{i,j}$ starts from the generation step $j \in \{0, c, \ldots, kc\}$ and completes $c$ steps actions based on the generated sub-goal $\mathbf{z}_j^i$.

### 4.3 Decision Mamba-Hybrid with Valuable Sub-goals

DM-H links Mamba and the transformer through the sub-goal $\mathbf{z}$ to efficiently recall long-term contexts while ensuring high-quality predictions. As sub-goals are commonly hidden in the offline data, Mamba must infer them along with the transformer training. However, it remains to be seen whether the transformer will benefit from $\mathbf{z}$ or ignore it and only imitate expert behaviors based on the local context. Therefore, we select extra high-value states from the offline data and transform them into sub-goals $\mathbf{z}$ to align actions generated by the transformer.

**Valuable sub-goals.** Intuitively, a sub-goal should be highly valuable for the agent to reach and have a high probability of appearing at subsequent time steps of the current state in the trajectory. When reached sequentially, these states should mark milestones in the trajectory, making it highly probable that the agent successfully performs the task. In this point, the valuable sub-goals guide the agent through the task, meaning they have the same purpose as the returns-to-go in the Decision Transformer [5]. Therefore, we can model this behavior by finding states with high accumulated reward values in the trajectory. Specifically, for state $s_i$ at timestep $i$, the value of state $s_j$ at timestep $j$ is $\sum_{t=i+1}^{j} r_k$, where $i + 1 \leq j \leq T$. However, this may prioritize the last state of trajectories when the environment only provides positive rewards. To encourage selecting states that are close to the current state $s_i$, we divide the accumulated rewards by the distance between their timesteps $\sum_{t=i+1}^{j} r_k / (j - i)$. With the punishment of the distance, the weighted average of accumulated rewards can identify the short-term and important future states.

Based on the selected sub-goals, we reconstruct the local short-term sequence (Equation (6)) by replacing $\mathbf{z}$ generated from Mamba. The reconstructed local short-term sequence is represented as:

$$\tau_g^{i,j} = (f(s_g^i), s_j^i, a_j^i, r_j^i, f(s_g^i), s_{j+1}^i, a_{j+1}^i, r_{j+1}^i, \ldots, f(s_g^i), s_{j+c-1}^i, a_{j+c-1}^i, r_{j+c-1}^i), \quad (7)$$

where $s_g^i$ is the most valuable sub-goal for state $s_j^i$ and $f$ is a linear layer that maps $s_g^i$ to the same dimension of $\mathbf{z}_j^i$. The reconstructed local short-term sequence aligns the actions generated by the transformer with subsequent high-valued states. Since Equation (6) is consistent with the reconstructed sequence except for $\mathbf{z}_j^i$, this encourages Mamba module to generate high-value sub-goals from the long-term context to ensure that the transformer module predicts similar actions.

### 4.4 Implementation of DM-H

**Architecture.** We feed $n$ trajectories into Mamba module, which results in $3 \times n \times T/c$ tokens, with one token for each of the three modalities: state, reward, and completion. In the transformer module, we feed $4 \times c$ tokens, with one token for each of the four modalities: sub-goal, state, action, and reward. To create the token embeddings, we train a linear layer for each modality, which transforms the raw inputs into the desired embedding dimension, followed by layer normalization [2]. Finally, we freeze a linear layer that maps the high-value sub-goals $s_g$ in Equation (7) to the same dimensions as the sub-goals $\mathbf{z}$ generated by Mamba.

**Offline Training and Online Testing.** During offline training, we are given a dataset of offline trajectories, where the trajectories can be suboptimal. In each iteration, we sample minibatches of trajectories from the dataset. Then, Mamba module first predicts the sub-goals $\mathbf{z}_t$ every $c$ steps, given the input token $s_t$ and past trajectories. Then, the transformer module autoregressively predicts $c$ steps of actions $\{a_t, \ldots, a_{t+c-1}\}$ given $\mathbf{z}_t$ and $\{s_t, \ldots, s_{t+c-1}\}$. Meanwhile, we use the weighted average of accumulated rewards to select valuable sub-goals from the offline data and feed them to the transformer model to predict the same $c$ steps actions. The predicted actions are evaluated with either cross-entropy loss or mean-squared error, depending on whether the actions are discrete or continuous. The losses from each step are averaged and updated in all modules end-to-end. At online testing, we roll out the DM-H with multiple trajectories and report the return of the last trajectory. Following the configuration from related works [20, 29], we set a context size across $n = 4$ episodes. The pseudocode for DM-H is summarized in Appendix A. Source code and more hyperparameters are described in Appendix B.

## 5 Experiments

In this section, we will introduce datasets and baselines in Section 5.1. Then, in Section 5.2, Section 5.3, Section 5.4, and 5.5, we report the comparison results, ablation study, and parameters sensitivity analysis. In Appendix C, we report additional results about offline training time, online testing time, and ablation study.

### 5.1 Environmental Settings

**Dataset: Grid World.** We first consider the discrete control environments from the Grid World [31], which is a commonly used benchmark for recent in-context RL methods. The environments support many tasks that cannot be solved through zero-shot generalization after pre-training because

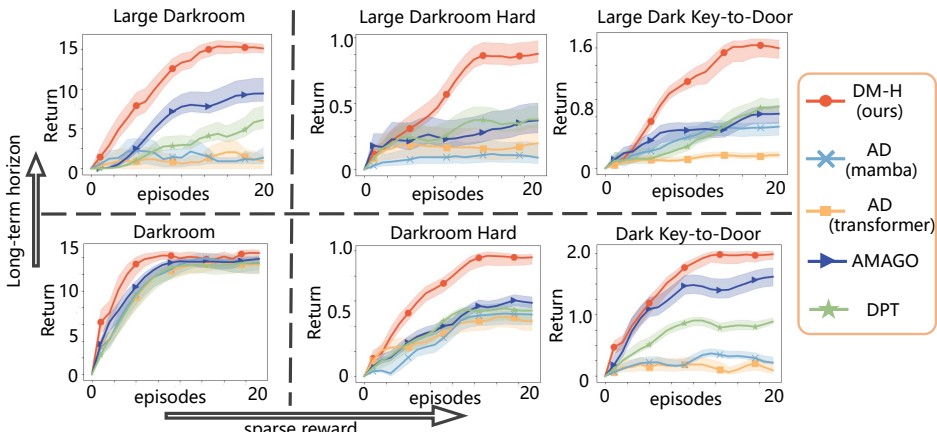

Figure 2: Results for Grid World. An agent is expected to solve a new task by interacting with the environments for 20 episodes without online model updates. Our DM-H significantly outperforms baselines on long-term tasks with sparse rewards because it inherits the merits of transformers and Mamba in high-quality prediction and long-term memory.

these tasks cannot be inferred easily from the observation. Specifically, we test our method on three environments: Darkroom, Darkroom Hard, and Dark Key-to-Door. In addition, we create a long-term variant of Large Darkroom, Large Darkroom Hard, and Large Darkroom Key-to-Door, where the coordinate space of each environment is expanded to 20 times, and the episode length is expanded 10 times. The dataset is collected from learning histories that are generated by training gradient-based RL algorithms, such as Deep Q-Network [34]. For each environment, we randomly create 60 tasks from the coordinate space and collect data for 1 million steps.

**Dataset: Tmaze.** We also evaluate our method on the Tmaze [35], a benchmark for testing the recall ability of in-context agents. In Tmaze, any policy that achieves the maximum return must be able to recall information from the first step at the final step. Since the task horizon can be set arbitrarily, it is often used to test the limits of the model's processing of long-term memory.

**Dataset: D4RL.** D4RL [13] is a commonly used offline RL benchmark, including continuous control tasks. The dataset is collected from Mujoco environments, including HalfCheetah, Hopper, and Walker. The episode length in D4RL is 1000, which is far more than that of Grid World. Therefore, current in-context RL methods require huge computational costs in D4RL, even though it is a commonly used benchmark for conventional RL algorithms.

**Baselines.** We investigate the effectiveness and efficiency of DM-H relative to in-context RL, dedicated offline RL, and imitation learning algorithms. Our baselines can be categorized as follows:

- In-context RL: These methods use the transformer to model trajectory sequences and predict actions autoregressively. We compare with recent methods, AMAGO [16], Decision Pretrained Transformer (DPT) [30], and Algorithm Distillation (AD) [29], which achieve impressive results based on the setting of across-episodic contexts.
- Temporal-difference learning: Most temporal-difference (TD) learning methods use an action space constraint or value pessimism and will serve as faithful comparisons to DM-H, representing standard RL methods. Following recent work [20], we consider state-of-the-art TD3+BC [14] that is demonstrated to be effective on D4RL.
- Imitation learning: Imitation learning methods similarly utilize supervised losses for training, such as Behavior Cloning (BC) [42] and Decision Transformer (DT) [5]. We compare with BC-10%, which is shown to be competitive with state-of-the-art on D4RL. DT also uses a transformer to predict actions autoregressively but is limited to a single episode context.

For all comparison methods, we adhere closely to the original hyper-parameter settings. To evaluate DM-H and other in-context RL algorithms, we roll out 10 episodes in D4RL and 20 episodes in Grid World and Tmaze. For each result, we report mean and standard error across 10 random seeds.

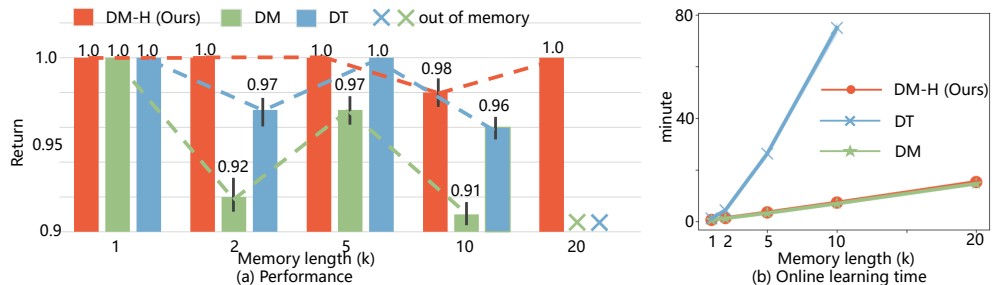

Figure 3: Results for (a) performance and (b) online testing times on Tmaze tasks. We train each method to address Tmaze tasks that have different horizons until we run out of GPU memory at context length to achieve 10k (DT, DM) or 20k (our DM-H). We report the online testing time for 20 episodes of Tmaze tasks.

Table 2: Results for D4RL datasets. DM-H outperforms both in-context RL (DT, AD) and supervised learning (BC) and performs competitively with conventional RL algorithms (TD3+BC and TD3) on almost all tasks.

| Dataset | Environment | BC-10% | TD3+BC | TD3 | DT | AD (Mamba) | AD (Transformer) | DM-H |
|---|---|---|---|---|---|---|---|---|
| Med-Expert | HalfCheetah | 94.11 | **96.59** | 87.60 | 93.40 | $92.21_{\pm 0.60}$ | $94.21_{\pm 0.46}$ | $96.21_{\pm 0.28}$ |
| Med-Expert | Hopper | 113.13 | 113.22 | 98.41 | 111.18 | $110.82_{\pm 0.56}$ | $108.32_{\pm 0.95}$ | $\mathbf{117.19}_{\pm 0.65}$ |
| Med-Expert | Walker2d | 109.90 | 112.21 | 100.52 | 108.71 | $108.31_{\pm 0.52}$ | $111.36_{\pm 0.46}$ | $\mathbf{118.21}_{\pm 0.56}$ |
| Med | HalfCheetah | 43.90 | **48.93** | 34.60 | 42.73 | $41.92_{\pm 0.11}$ | $42.28_{\pm 1.18}$ | $45.45_{\pm 0.35}$ |
| Med | Hopper | 73.84 | 70.44 | 56.98 | 69.42 | $73.65_{\pm 1.23}$ | $72.58_{\pm 0.54}$ | $\mathbf{83.15}_{\pm 0.63}$ |
| Med | Walker2d | 82.05 | 86.91 | 70.95 | 74.70 | $78.26_{\pm 0.55}$ | $85.96_{\pm 0.46}$ | $\mathbf{88.29}_{\pm 0.76}$ |
| Med-Replay | HalfCheetah | 42.27 | 45.84 | 38.81 | 40.31 | $39.68_{\pm 0.13}$ | $41.28_{\pm 0.21}$ | $\mathbf{45.26}_{\pm 0.43}$ |
| Med-Replay | Hopper | 90.57 | 98.12 | 78.90 | 88.74 | $82.65_{\pm 1.15}$ | $91.32_{\pm 0.66}$ | $\mathbf{98.36}_{\pm 0.51}$ |
| Med-Replay | Walker2d | 76.09 | 91.17 | 65.94 | 71.22 | $70.92_{\pm 1.21}$ | $89.21_{\pm 1.42}$ | $\mathbf{95.66}_{\pm 1.16}$ |
| Total Average | | $80.65_{\pm 1.34}$ | $84.83_{\pm 1.10}$ | $70.28_{\pm 1.20}$ | $77.69_{\pm 1.45}$ | $76.97_{\pm 0.67}$ | $82.84_{\pm 0.70}$ | $\mathbf{87.53}_{\pm 0.59}$ |

## 5.2 Grid World Results

To evaluate DM-H's self-improvement capabilities in unseen tasks, we compared recent in-context RL methods in the Grid World environments. The agent is required to solve an unseen task by interacting with the environments for 20 episodes without online model updates. In addition, we added a variant of the representative in-context RL AD, replacing the transformer with Mamba as the backbone.

As shown in Figure 2, DM-H achieves state-of-the-art performance in a wide range of tasks. DM-H achieves $28\times$ times faster than baselines and more than double the effectiveness as the task horizon increases and rewards become sparse. In long-term tasks, DM-H leverages Mamba's ability to obtain long-term memories while retaining high-quality predictions established by the transformer in decision-making. On the contrary, AD (transformer) and AD (Mamba) only retain one aspect of the advantages of DM-H. In terms of sparse rewards, DM-H is similar to the structure of hierarchical RL, in which Mamba performs a decision every $c$ steps and is, therefore, better at handling reward sparse scenarios. Overall, DM-H demonstrated that the hybrid method is not only feasible but combines more advantages synergetically.

## 5.3 Tmaze Results

In this section, we test the limits of DM-H's recall capabilities in the Tmaze task. Solving this task requires accurate recall of the first step at the last step. Therefore, we set the context length equal to the task horizon and compare it with DT and DM, where DM replaces the DT's transformer backbone with a Mamba backbone. We train each method to recover the optimal policy until we run out of GPU memory at a context size equal to 20k (DM-H) or 10k (DT and DM).

As shown in Figure 3, DM-H achieves the maximum reward with minimum online testing costs at any task horizon, demonstrating that Mamba's recalled memory positively prompts the transformer. Regarding efficiency, DM-H is comparable to DM using only Mamba during online testing and outperforms DM during offline training (Figure 5 in Appendix C). This is because (1) the context of

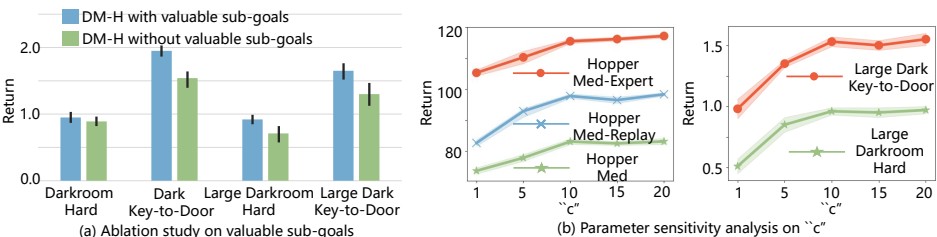

Figure 4: (a) The ablation study on DM-H with or without valuable sub-goals. (b) The parameter sensitivity analysis of "$c$."

the transformer in DM-H is fixed to the hyperparameter $c$, which does not change with the task length. (2) Mamba model generates a sub-goal every $c$ steps, thereby shortens the sequence it processes by $c\times$ times. Benefiting from these merits, DM-H handles tasks twice the length and has fewer memory resource requirements than DT and DM.

## 5.4 D4RL Results

In addition to navigation tasks, we also test DM-H on the control tasks from the D4RL dataset, which is commonly used in conventional offline RL methods. Based on previous work [13], the results on D4RL are normalized so that 100 denotes an expert policy. Baseline numbers are reported by the Agentic Transformer [20] and from the D4RL paper. As shown in Table 2, DM-H outperforms baselines in a majority of the tasks and is competitive with the state-of-the-art in the remaining tasks. In the TD learning and imitation learning categories, TD3+BC is generally the most remarkable algorithm. Compared with them, the superior performance demonstrates the self-improvement of DM-H on suboptimal data.

## 5.5 Ablation Study and Parameter Sensitivity Analysis

**Ablation Study on Valuable Sub-goals.** To validate the effectiveness of valuable sub-goals, we conduct an ablation study of DM-H in Grid World tasks. Figure 4 (a) presents the ablation results, which report the mean episode return across 10 seeds. We can observe that sub-goals can significantly improve DM-H's performance, proving that the sub-goal strengthens the transformer's dependency on long-term contexts from Mamba. In Appendix C, we also carry out ablation studies in the D4RL dataset. Please refer to Figure 6.

**Sensitivity of Key Hyperparameters.** In this experiment, we introduce an important hyperparameter $c$. A large $c$ enables Mamba model to scan multiple episodes with smaller context sizes, significantly improving the computational efficiency. In addition, $c$ also controls the short-term context length in the transformer, which affects the quality of generated actions. As shown in Figure 4 (b), DM-H performs well within the appropriate range of $c$, i.e., $10 \leq c \leq 20$.

## 6 Conclusion, Limitations, and Broader Impacts

In this work, we propose an in-context RL method DM-H that achieves both high effectiveness and efficiency in long-term tasks. The core idea of DM-H is to use sub-goal settings to leverage Mamba's ability to efficiently recall long sequence contexts while using the transformer to perform high-quality predictions. Unlike current in-context RL methods limited to short-term tasks, DM-H is also good at standard RL benchmarks, which typically have long-term sequences under the across-episodic context setting. On the Grid World, D4RL, and Tmaze benchmarks, we demonstrate that DM-H can outperform baselines in both efficiency and effectiveness.

Regarding limitations, our method has an important hyperparameter, which is the context length of the transformer "$c$." As "$c$" increases, Mamba model can scan multiple episodes with smaller context sizes, significantly improving the computational efficiency. On the contrary, as "$c$" decreases, Mamba can generate high-quality sub-goals from the context that is closer to the original episode. However, this reduces the short-term context available to the transformer when making predictions. Therefore, a meaningful future direction is for "$c$" to adapt autonomously to different tasks. In terms of the

potential broader impact, we do not anticipate any negative ethical and societal impacts of our work while using our method in practice.

## Acknowledgments

This work was supported by the National Key R&D Program of China under Grant No. 2021ZD0112500; the National Natural Science Foundation of China under Grant Nos. U22A2098, U2341229, 62172185, 61976102, 62206105, 62202200 and 62476110; the Key R&D Project of Jilin Province under Grant Nos. 20240212003GX and 20240304200SF; the International Cooperation Project of Jilin Province under Grant No. 20220402009GH.

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

*Appendix of paper "Decision Mamba: Reinforcement Learning via Hybrid Selective Sequence Modeling"*

## A  Pseudocode of Decision Mamba-Hybrid

---

**Algorithm 1:** Decision Mamba-Hybrid.

---

**Input:** A dataset of Trajectories, Max Iterations $M$ as training phase, Max episodes $m$ at testing phase, A number of trajectories $n$ in across-episodic contexts used in Mamba model, A number of steps of actions $c$ for one sub-goals

**Output:** The generated actions

1  **//Training**
2  **for** $i = 1$ **to** $M$ **do**
3      Randomly sample $n$ episodes from dataset $s = (\tau^1, \tau^2, \ldots, \tau^n)$
4      Sort $n$ episodes ascending according to their returns $\sum_{t=0}^{T} r_t^1 \le \sum_{t=0}^{T} r_t^2 \le \cdots \le \sum_{t=0}^{T} r_t^n$
5      Concatenate $n$ episodes as a across-episodic context $s_h = (\tau_m^1, \tau_m^2, \ldots, \tau_m^n)$ based on Equation (5)
6      Mamba module predicts the next sub-goals tokens from the across-episodic context
7      Build a local short-term sequence every $c$ steps based on Equation (6)
8      Select the high-value states from the offline data based on the weighted average of accumulated rewards $\sum_{t=i+1}^{j} r_k / (j - i)$ Build a similar local short-term sequence every $c$ steps based on Equation (7)
9      The transformer module predicts the next $c$ steps action tokens for each predicted sub-goals token from Mamba and selected high-value sub-goals
10     Train Mamba and transformer models based on the loss of predicted actions end-to-end
11 **end**
12 **//Testing**
13 **for** $i = 1$ **to** $m$ **do**
14     Start a new episode $i$ and reset the timestep $t = 0$
15     **while** $t \le T$ **do**
16        Mamba model generates next sub-goal token $\mathbf{z}_t^i$ based on the historical trajectories $(\tau_m^1, \tau_m^2, \tau_m^{i-1}, \ldots, s_t^i)$, where $\tau_m^i$ is expressed as Equation (5)
17        **for** $k = 0$ **to** $c - 1$ **do**
18           The transformer model generates next action $a_{t+k}^i$ based on the local context $(\mathbf{z}_t^i, s_t^i, a_t^i, r_t^i, \ldots, \mathbf{z}_t^i, s_{t+k}^i)$
19        **end**
20        Compute the sum of $c$ steps rewards $\hat{r}_t^i = \sum_{k=t}^{t+c-1} r_k^i$
21        Receive the next observation or state $s_{t+c}^i$
22        Update the across-episodic context $(\tau_m^1, \tau_m^2, \tau_m^{i-1}, \ldots, s_t^i, \mathbf{z}_t^i, \hat{r}_t^i, d_t^i, s_{t+c}^i)$
23        Update time step $t = t + c$
24     **end**
25 **end**

---

In Algorithm 1, we introduce the training and testing process of DM-H. At each iteration, we first construct a long-term sequence consisting of multiple trajectories, as described in lines 3-5. Mamba model predicts sub-goals based on the long-term sequence (line 6). Then, each sub-goal will correspond to a short sequence of $c$ steps actions, as described in line 7. To encourage the transformer to rely on Mamba's predictions, we select the high-value states from the offline data and transform them into sub-goals to reconstruct another short sequence of $c$ steps actions (line 8). Based on the short sequence and reconstructed sequences, the transformer model predicts $c$ steps actions (line 9). Finally, the predicted actions are evaluated with either cross-entropy loss or mean-squared error, depending on whether the actions are discrete or continuous. The losses from each time step are averaged and updated in all models end-to-end, as described in line 10.

During online testing, DM-H needs to generate actions autoregressively and interact with the environment in $m$ episodes. At step $t$ of episode $i$ (line 16), Mamba module first generates a sub-goal token $\mathbf{z}_t^i$

Table 3: Hyperparameters of DM-H.

| | Hyperparameters | Value |
|---|---|---|
| Mamba | Number of layers | 2 |
| | Embedding dimension | 128 |
| | Expand factor | 2 |
| | Convolution size | 4 |
| Transformer | Number of layers | 3 |
| | Number of attention heads | 3 |
| | Embedding dimension | 128 |
| | Activation function | ReLU |
| | $c$ steps controlled by one sub-goal | 20 D4RL and Large Grid World |
| | | 5 Grid World and Tmaze |
| Training | Batch size | 128 |
| | Dropout | 0.1 |
| | Learning rate | 1e-4 |
| | Learning rate decay | Linear warmup for 1e5 steps |
| | Grad norm clip | 0.25 |
| | Weight decay | 1e-4 |
| | Number of trajectories to form across-episodic contexts $n$ | 4 (Large) Dark Key-to-Door |
| | | 10 other tasks in Grid World |
| | | 4 D4RL |
| Testing | Target return for Tmaze | 1 |
| | Target return for HalfCheetah | 12000 |
| | Target return for Hopper | 3600 |
| | Target return for Walker | 5000 |
| | Target return for Darkroom | 20 |
| | Target return for Darkroom Hard | 1 |
| | Target return for Darkroom Key-to-Door | 2 |
| | Target return for Large Darkroom | 15 |
| | Target return for Large Darkroom Hard | 1 |
| | Target return for Large Darkroom Key-to-Door | 2 |
| | Number of trajectories to form across-episodic contexts $n$ | 4 (Large) Dark Key-to-Door |
| | | 10 other tasks in Grid World |
| | | 4 D4RL |

conditioned on the historical context $(\tau_m^1, \tau_m^2, \tau_m^{i-1}, \ldots, s_t^i)$, where $\tau_m^i$ is expressed as Equation (5). Then, the transformer will generate the following $c$ steps actions $(a_t, \ldots, a_{t+c-1})$ autoregressively, as described in lines 17-19. Finally, we update the across-episodic context for generating the next sub-goal $\mathbf{z}_{t+c}^i$, as described in lines 20-23.

## B  Experimental Details

**Dataset: Grid World.**  The evaluation environments of Grid World provide a 2D discrete POMDP where an agent spawns in a room and must find a goal location. The agent only observes its own $(x, y)$ coordinates but does not know the goal location, which is required to deduce it from the rewards received. The room dimensions are $9 \times 9$ with the agent's possible actions, including moving one step either left, right, up, down, or staying idle. In Darkroom, an episode lasts 20 steps, and the agent can obtain a reward ($r = 1$) each time the goal is achieved. The Darkroom Hard is a variant of Darkroom. In the Darkroom Hard, agents only obtain a reward when the goal is achieved first. In the Dark Key-to-Door, the length of an episode is 50, where the agent is required to locate an invisible key to receive a one-time reward first and then identify an invisible door to obtain another one-time reward. In addition, we create a long-term variant of Large Darkroom, Large Darkroom Hard, and Large Darkroom Key-to-Door, where the coordinate space of each environment is expanded to $40 \times 40$, and the episode length is expanded 10 times.

**Dataset: D4RL.**  D4RL [13] is a commonly used offline RL benchmark, including continuous control tasks. The different dataset settings are described below.

• Medium: 1 million timesteps generated by a "medium" policy that performs approximately one-third as well as an expert policy.

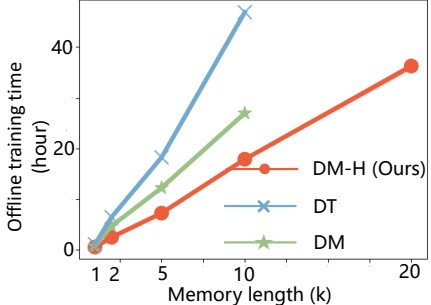

Figure 5: Results for offline training times on Tmaze tasks. We train each method to address Tmaze tasks that have different horizons until we run out of GPU memory at context length to achieve 10k (DT, DM) or 20k (DM-H). We report the training times for 10k gradient updates on Tmaze tasks.

Table 4: Results for offline training and online testing times. We report the offline training time per 10k gradient updates, the online testing time for 20 episodes over Grid World, and 10 episodes over D4RL. As the task length increases, the context length is forced to grow exponentially, resulting in a square increase in computational costs. In contrast, DM-H completes trial-and-error on Mamba model in sizes smaller than other baselines, significantly reducing computational costs.

| Context size (step) | Tasks | Offline training (hour) | | | Online testing (minute) | | |
|---|---|---|---|---|---|---|---|
| | | AD (Mamba) | AD (Transformer) | DM-H (Ours) | AD (Mamba) | AD (Transformer) | DM-H (Ours) |
| 200 | Darkroom | 0.21 | 0.23 | 0.18 | 0.20 | 0.61 | 0.21 |
| | Darkroom Hard | 0.22 | 0.28 | 0.20 | 0.20 | 0.56 | 0.19 |
| | Darkroom Dynamic | 0.24 | 0.31 | 0.21 | 0.21 | 0.62 | 0.22 |
| | Dark Key-to-Door | 0.65 | 1.01 | 0.41 | 0.45 | 1.50 | 0.46 |
| 2000 | Large Darkroom | 3.52 | 4.70 | 2.38 | 5.06 | 45.08 (9×) | 5.12 |
| | Large Darkroom Hard | 4.26 | 6.69 | 2.78 | 5.61 | 44.96 (7×) | 5.59 |
| | Large Darkroom Dynamic | 2.71 | 5.84 | 2.63 | 5.21 | 42.12 (8×) | 5.36 |
| | Large Dark Key-to-Door | 6.87 | 18.23 | 3.16 | 5.88 | 76.79 (12×) | 6.08 |
| 4000 | HalfCheetah | 28.56 | 37.10 | 20.96 | 6.16 | 173.11 (28×) | 6.21 |
| | Walker2d | 26.25 | 33.77 | 19.96 | 6.01 | 172.34 (26×) | 6.11 |
| | Hopper | 18.15 | 22.23 | 11.52 | 6.05 | 172.92 (26×) | 6.06 |

- Medium-Replay: 1 million timesteps collected from the replay buffer of an agent trained to the performance of a "medium" policy.

- Medium-Expert: It consists of 1 million timesteps generated by the "medium" policy and another 1 million timesteps generated by the expert policy.

**Dataset: Tmaze.** The Tmaze task provides a T-shaped maze where the agent is rewarded only by reaching the end point at the last step. However, the endpoint's location is unknown; it is only told once, at an oracle early in the maze. Therefore, any policy that achieves the maximum return must be able to recall information from the first step at the final step.

**Compute.** Experiments are carried out on NVIDIA GeForce RTX 3090 GPUs and NVIDIA A10 GPUs. Besides, the CPU type is Intel(R) Xeon(R) Gold 6230 CPU @ 2.10GHz. In particular, our method has lower memory requirements because it naturally shortens the across-episodic contexts.

**Hyperparameters.** The default length of across-episodic is four trajectories unless mentioned otherwise. In D4RL, Tmaze, and Large Grid World, the transformer model generates $c = 20$ steps actions while Mamba model generates one sub-goal. In conventional Grid World, we set $c = 5$ because the task is too short. In summary, Table 3 shows the hyperparameters used in our DM-H model.

## C  Additional Experimental Results

**Additional Tmaze Results.** As described in the experiments section, we use Tmaze tasks to test the limits of DM-H's recall capabilities. We train each method to recover the optimal policy until

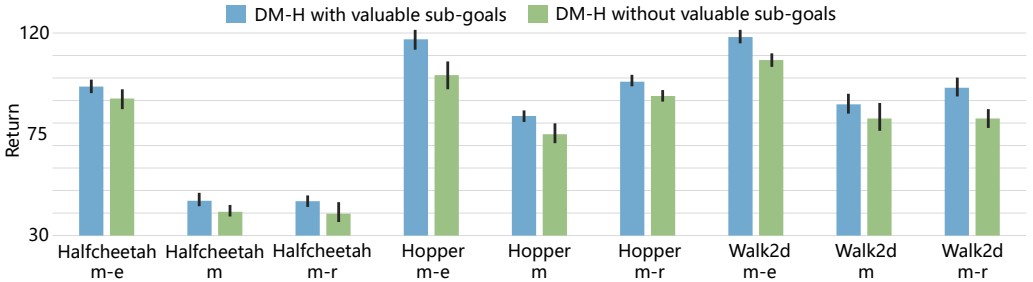

Figure 6: The ablation study on DM-H with or without valuable sub-goals.

we run out of GPU memory at a context size equal to 20k (DM-H) or 10k (DT and DM). As shown in Figure 5, DM-H achieves the maximum reward with minimum offline training costs at any task horizon. Regarding efficiency, DM-H is even faster than DM using only Mamba. This is because (1) the context of the transformer in DM-H is fixed to the hyperparameter $c$, which does not change with the task length. (2) Mamba model generates a sub-goal every $c$ steps, which shortens the sequence it processes by $c\times$ times.

**Additional Evaluation of Computing Costs.** An important property of in-context RL is that it can improve itself without expensive gradient updates during online testing. However, the computational costs of forward propagation are hidden in short-horizon tasks. Therefore, we reported the offline training time per 10k gradient updates, the online time for 20 episodes over Grid World, and 10 episodes over D4RL. As shown in Table 4, our DM-H has efficient training and significantly reduces the online testing time compared to the baselines, approximately **28×** times faster in D4RL and **12×** times faster in large Grid World. As the task length increases, the online testing time of AD grows quadratically. This is because the across-episodic contexts multiply the sequence length, leading to intolerable computational costs in the self-attention mechanism. In contrast, DM-H leverage Mamba to process long-term sequence, where the computational complexity increases linearly with length.

**Additional Ablation Study on Valuable Sub-goals.** To validate the effectiveness of valuable sub-goals, we also conduct an ablation study of DM-H in D4RL tasks. Figure 6 presents the ablation results, which report the mean episode return across 10 seeds. We can observe that sub-goals can significantly improve DM-H's performance, proving that the sub-goal strengthens the transformer's dependency on long-term contexts from Mamba.

