# OpenReview forum: "Decision Mamba: Reinforcement Learning via Hybrid Selective Sequence Modeling"
_NeurIPS.cc/2024/Conference — NeurIPS 2024 poster_

### Official Review · Reviewer_UASH · 2024-07-03

**Soundness:** 4
**Presentation:** 4
**Contribution:** 4
**Rating:** 10
**Confidence:** 5

**Summary:**

This paper investigates an emerging foundation model, Mamba, in Reinforcement Learning (RL) scenarios and compares it with Transformer in terms of effectiveness and efficiency. The authors find that in-context RL methods with Mamba as the backbone are generally more efficient than Transformer, but there is no significant improvement in effectiveness. Then, this paper proposes a Hybrid Mamba (HM) with the merits of transformers and Mamba in high-quality prediction and long-term memory. Finally, this paper conducts experiments on three benchmarks to exhibit its improved effectiveness and efficiency.

**Strengths:**

1.	The paper is commendably well-written and coherent, effectively explaining complex ideas in an accessible manner. The authors explored the potential of the widely discussed model Mamba in the context of RL and compared it with Transformer in terms of effectiveness and efficiency.
2.	The authors proposed a novel hybrid model that inherits the merits of both Transformer and Mamba in a goal-conditional manner. The main advantage of using the hybrid structure is that when the time horizon is very long, as in the D4RL tasks, several episodes/trials are required for good in-context learning, as in the larger Dark Room and Tmaze environments.
3.	HM improves training and inference speed by reducing the horizon of the transformer model. This can be particularly important in applications such as robotics, which require high-frequency control.
4.	The experimental evaluation, meticulously designed to include several baselines and diverse tasks, demonstrates the algorithm's strengths.

**Weaknesses:**

1.	The baseline AD (Mamba) in Figure 2 and the baseline DM in Figure 3, which appear to be AD (Transformer) and DT variants, are crucial for the readers' understanding of how Mamba replaces the Transformer architecture. However, the lack of explanation of these two baselines in the experimental setup section might confuse readers.
2.	Some experimental settings are not explained clearly. In Section 5.3, the authors do not explain what GPU device they used. Although the device is introduced in Appendix A, it is recommended that it be explained clearly in the main text.

**Questions:**

1.	In Table 1 and Table 2, the author used AD (Mamba) as the primary baseline. However, in Figure 3, the author used the DM baseline instead. What is the main difference between AD (Mamba) and DM?
2.	The experiments demonstrated that the online testing of HM in the long-term task is 28 times faster than the transformer-based method. However, can this hybrid structure also inherit Mamba's high efficiency in terms of training cost?

**Limitations:**

The author discusses limitations and potential negative societal impacts in Section 6.

---

> ### Author Rebuttal · Authors · 2024-08-06
>
> We are particularly encouraged that Reviewer UASH finds our method effective.
>
> ### [I].Reply to the weakness
> >**[1/2]W1. The baseline AD (Mamba) in Figure 2 and the baseline DM in Figure 3, which appear to be AD (Transformer) and DT variants, are crucial for the readers' understanding of how Mamba replaces the Transformer architecture. However, the lack of explanation of these two baselines in the experimental setup section might confuse readers.**
>
> Thanks for the reviewer's suggestions. The difference between AD (Mamba) and DM is whether the context covers multiple trajectories. We highlight the advantages and disadvantages of Mamba by comparing it with two transformer-based methods, AD (Transformer) and DT. In the revised manuscript, we will add more detailed instructions in Section 5.1 of the experiment.
>
> >**[2/2]W2. Some experimental settings are not explained clearly. In Section 5.3, the authors do not explain what GPU device they used. Although the device is introduced in Appendix A, it is recommended that it be explained clearly in the main text.**
>
> Experiments are carried out on NVIDIA GeForce RTX 3090 GPUs and NVIDIA A10 GPUs. Besides, the CPU type is Intel(R) Xeon(R) Gold 6230 CPU @ 2.10GHz. As suggested, we will emphasize the device in the revised manuscript.
>
> ### [II].Reply to the questions
> >**[1/2]Q1. In Table 1 and Table 2, the author used AD (Mamba) as the primary baseline. However, in Figure 3, the author used the DM baseline instead. What is the main difference between AD (Mamba) and DM?**
>
> The main difference between AD (Mamba) and DM is whether the context covers multiple trajectories. Since the Tmaze requires the policy to recall the first step at the current trajectory, we show the DT and DM comparison results in Figure 3. In contrast, the Grid world and d4rl tasks require the policy to improve itself based on the historical trajectories. Thus, we use AD (Mamba) as the primary baseline.
>
> >**[2/2]Q2. The experiments demonstrated that the online testing of HM in the long-term task is 28 times faster than the transformer-based method. However, can this hybrid structure also inherit Mamba's high efficiency in terms of training cost?**
>
> Yes, we are glad that the reviewer found our method also highly efficient in terms of training cost. Assume the sequence length is $n$. The computational complexity of the transformer is $O(n^2)$. In HM, the sequence is divided into a hierarchical structure. In the Mamba level, the sequence length is $\frac{n}{c}$, and the complexity is $O(\frac{n}{c})$, where the hyperparameter $c$ denotes the timestep interval at which each sub-goal is set to guide the transformer. In the transformer level, the sequence is divided into $\frac{n}{c}$ subsequences of length $c$, and the complexity becomes $O(\frac{n}{c}\cdot c^2)$. As the sequence length $n$ increases, the computational complexity of HM will be significantly lower than that of the transformer-based method, and thus, the training will be faster.

---

> > ### Author Response · Authors · 2024-08-12
> > **Looking forward to your reply**
> >
> > Dear Reviewer UASH,
> >
> > We value your positive feedbacks and constructive suggestions for our paper and sincerely appreciate your effort in reviewing it. We hope we have effectively addressed all the concerns raised. As the end of the discussion is approaching, we are wondering if there are any additional potential clarifications or suggestions that you think would help us improve this manuscript.
> >
> > Thank you again for your dedicated review and invaluable insights.
> >
> > Kind regards,
> >
> > Paper5823 Authors

---

> > > ### Comment · Reviewer_UASH · 2024-08-13
> > >
> > > Thanks for the author's clarification. I decided to keep my initial score. I rated this work highly for several reasons. First, Mamba is an emerging foundational model with significant potential, yet its performance in decision-making tasks has been underexplored. This paper fills that gap by systematically comparing Mamba's effectiveness and efficiency against the transformer model on classic RL tasks. Second, the authors propose a hybrid framework that leverages the strengths of both Mamba and transformer models in decision-making tasks. Overall, I believe this work will generate considerable interest within the NeurIPS community and advance the application of foundational models in RL.

---

### Official Review · Reviewer_YUg4 · 2024-07-08

**Soundness:** 2
**Presentation:** 3
**Contribution:** 2
**Rating:** 4
**Confidence:** 4

**Summary:**

This paper presents Hybrid Mamba (HM), a method that combines the Mamba model and Transformer to enhance reinforcement learning (RL) performance. HM leverages Mamba to generate high-value sub-goals, which then condition the transformer, leading to significant improvements in online testing efficiency and task-solving capabilities.

**Strengths:**

1. The paper is well-written and clear to read.
2. HM significantly accelerates testing speed, achieving up to 28 times faster results than baseline methods.
3. HM demonstrates superior performance across various benchmarks, such as D4RL, Grid World, and Tmaze.

**Weaknesses:**

1. This paper claims to present a in-context RL approach. The motivation of this paper is concerned with the problems encountered with the no-gradient updates in-context approach (line 28), where the policy network does not require parameter updates. However, this paper uses a global update approach, which is closer to gradient-based and conditional-based offline RL. It seems to contradict the original intention of this paper.

2. HM benefits from using a powerful subgoal encoder (Mamba in this case) and conditioning the policy network with subgoals. The performance improvement is expected and unsurprising due to the advantages inherent in conditional-based RL algorithms. Hence, it is necessary to further explain the unique contributions of combining Mamba and causal transformer in this paper.

3. If the sub-goal encoder are replaced with other advanced and efficient sequence encoders (e.g., flash-attention1/2 [1,2], x-lstm [3]), would it also yield better or more efficient performance?

4. The experiments demonstrating HW's efficacy in capturing long-term dependencies are unconvincing. Achieving good results in tasks with an arbitrarily horizon (e.g., Tmaze) does not necessarily prove effective long-term memory embedding. It is crucial to test the stability and performance of HM with varying horizon lengths or other length-based settings. For example, Mamba’s original paper [4] demonstrated the ability to capture long-term dependencies through the scaling laws.

5. Could the authors clarify in which specific aspects HM's training time is faster than DT's? Since HM appears to be a combination of Mamba and DT.

6. There are parts of the paper that are not clearly explained. For instance, in lines 228-233, it is mentioned that the transformer predicts a c-step action sequence (named $a_1$ here) through the sub-goal $z_t$ and another c-step action sequence (named $a_2$) through valuable sub-goals from offline data. How are $a_1$ and $a_2$ subsequently updated or processed?

7. (minor) The paper contains some typos and inconsistencies in tense usage. For example, in the related work section, the section on Mamba uses the present tense, while the section on in-context RL uses the past tense. These should be corrected for consistency. In addition, what's the meaning of the different gaussian distribution figures in Figure 1?

*Reference:*

[1] Dao T, Fu D, Ermon S, et al. Flashattention: Fast and memory-efficient exact attention with io-awareness. NeurIPS 2022.

[2] Dao T. Flashattention-2: Faster attention with better parallelism and work partitioning. ICLR 2024.

[3] Beck M, Pöppel K, Spanring M, et al. xLSTM: Extended Long Short-Term Memory. arXiv 2024.

[4] Gu A, Dao T. Mamba: Linear-time sequence modeling with selective state spaces. arXiv 2023.

**Questions:**

Please see weakness. If I have misunderstood some parts of the paper, I welcome corrections and further discussion.

**Limitations:**

The authors raise some limitaions, for example, how to control the setting of hyperparameter $c$, which is not addressed in this paper but is claimed to be solved in the future.

---

> ### Author Rebuttal · Authors · 2024-08-06
>
> Thanks for the reviewer's positive appraisal, insightful comment, and criticism of our paper.
>
> ### [I].Reply to the Weakness
> >**[1/7]W1. This paper claims to present a in-context RL approach. The motivation of this paper is concerned with the problems encountered with the no-gradient updates in-context approach (line 28), where the policy network does not require parameter updates. However, this paper uses a global update approach, which is closer to gradient-based and conditional-based offline RL. It seems to contradict the original intention of this paper.**
>
> We want to clarify the setting of transformer-based in-context RL, which is divided into two phases: training and testing. In the training phase, in-context RL designs the architecture of policy models and trains the model by predicting the action tokens in carefully designed sequences, such as multiple trajectories in AD [1]. This process involves parameter updates and can be regarded as a kind of pre-training. In the testing phase, we deploy the model in a new task and can only make inferences in the context obtained by interacting with the environment without involving parameter updates.
>
> In the method section, we introduced the architecture of HM and how to construct sequences for training. Then, at the end of the section, we summarized its testing process. We roll out the HM with multiple trajectories and report the return of the last trajectory. In-context learning is reflected by the HM, which can predict better actions by recalling historical trajectories from the context. In Figure 2 in the manuscript, the ascending reward curves are achieved by learning in the contexts without requiring parameter updates.
>
>
> >**[2/7]W2. HM benefits from using a powerful subgoal encoder (Mamba in this case) and conditioning the policy network with subgoals. The performance improvement is expected and unsurprising due to the advantages inherent in conditional-based RL algorithms. Hence, it is necessary to further explain the unique contributions of combining Mamba and causal transformer in this paper.**
>
> Thanks for the reviewer's suggestion. Mamba is well known for its competitive effectiveness in transformer-based models while achieving linear complexity in long sequences. Its outstanding performance in NLP and visual tasks has attracted more and more attention. Therefore, our first contribution is to explore the potential of Mamba in decision-making tasks. In traditional RL tasks, we demonstrate that the Mamba model is more efficient but slightly inferior to the transformer model in terms of effectiveness. Therefore, we propose the HM method that inherits the merits of Mamba and transformers, achieving both high effectiveness and efficiency in long-term RL tasks. HM is inspired by the idea of conditional RL, proving that this hybrid structure is a promising way to leverage the strengths of Mamba and Transformer to complement their weaknesses.
>
> >**[3/7]W3. If the sub-goal encoder are replaced with other advanced and efficient sequence encoders (e.g., flash-attention1/2, x-lstm), would it also yield better or more efficient performance?**
>
> It is possible to use other sequence encoders to generate sub-goals, such as the reviewer's suggested flash-attention and x-lstm. The x-lstm has been proposed recently, and we did not find its open source code. Therefore, we tested HM by replacing the Mamba module with the flash-attention. Due to the limited time and device, we report the results of flash-attention1 across three random seeds.
>
> **Table 1. The performance of different sub-goal encoder.**
> |Sub-goal encoder|HalfCheetah-Med-Expert|HalfCheetah-Med|HalfCheetah-Med-Replay|Hopper-Med-Expert|Hopper-Med|Hopper-Med-Replay|Walker2d-Med-Expert|Walker2d-Med|Walker2d-Med-Replay|
> |-|-|-|-|-|-|-|-|-|-|
> |Mamba|96.12 $\pm$ 0.28|45.45 $\pm$ 0.35|45.26 $\pm$ 0.4|117.19 $\pm$ 0.65|83.15 $\pm$ 0.63|98.36 $\pm$ 0.51|118.21 $\pm$ 0.56|88.29 $\pm$ 0.76|95.66 $\pm$ 1.16|
> |flash-attention|94.91 $\pm$ 0.18|44.22 $\pm$ 0.16|43.98 $\pm$ 0.35|116.09 $\pm$ 0.68|81.58 $\pm$ 0.33|96.85 $\pm$ 0.64|117.19 $\pm$ 0.49|86.26 $\pm$ 0.61|94.12 $\pm$ 0.88|
>
> The results show that flash attention is slightly inferior to Mamba on d4rl tasks. This is because the process of predicting sub-goals is non-autoregressive. As we clarified in the manuscript, the sub-goal is represented by a vector that is not explicitly present in the training data. This non-autoregressive process may not fully exploit the power of flash-attention. On the contrary, although Mamba was also first proposed for NLP tasks, its structure is closer to RNN, a commonly used architecture in RL. Similarly, X-LSTM should yield efficient performance because it is also an RNN-like model.
>
> >**[4/7]W4.The experiments demonstrating HW's efficacy in capturing long-term dependencies are unconvincing. Achieving good results in tasks with an arbitrarily horizon (e.g., Tmaze) does not necessarily prove effective long-term memory embedding. It is crucial to test the stability and performance of HM with varying horizon lengths or other length-based settings.**
>
> There is a misunderstanding about our testing of HM recalling long-term memory. We followed the setting of the previous in-context RL method AMAGO [2] to extend the horizon of tasks until we ran out of GPU memory. The horizon of HM also varies with the tasks from short-horizon to long-horizon. Although unlike indicators such as perplexity or accuracy in NLP tasks, the cumulative reward (return) in RL tasks has a similar meaning and is used to evaluate the model's performance. Figure 3(a) results in the manuscript show that HM with varying horizon lengths are stable and high-performance. In summary, we want to show that HM can effectively handle tasks that require recalling long-term memory, as tested by previous methods.
>
> Due to word limitations, we will answer the remaining questions in the next comment.

---

> ### Author Response · Authors · 2024-08-06
> **Rebuttal of the remaining questions**
>
> Due to word limitations, we answer the remaining questions in this comment. Please review this response after the rebuttal part.
>
> ### [I].Reply to the Weakness
>
> >**[5/7]W5.Could the authors clarify in which specific aspects HM's training time is faster than DT's? Since HM appears to be a combination of Mamba and DT.**
>
> Thanks for the reviewer's suggestion. Although HM combines Mamba and DT, its sequence length is the same as that of DT. Assume the sequence length is $n$. The computational complexity of DT is $O(n^2)$. In HM, the sequence is divided into a hierarchical structure. In the Mamba level, the sequence length is $\frac{n}{c}$, and the complexity is $O(\frac{n}{c})$, where the hyperparameter $c$ denotes the timestep interval at which each sub-goal is set to guide the DT. In the DT level, the sequence is divided into $\frac{n}{c}$ subsequences of length $c$, and the complexity becomes $O(\frac{n}{c}\cdot c^2)$. As the sequence length $n$ increases, the computational complexity of HM will be significantly lower than that of DT, and thus, the training will be faster. We will incorporate the above complexity analysis into the revised manuscript.
>
> >**[6/7]W6.There are parts of the paper that are not clearly explained. For instance, in lines 228-233, it is mentioned that the transformer predicts a c-step action sequence (named $a_1$ here) through the sub-goal $z_t$ and another c-step action sequence (named $a_2$) through valuable sub-goals from offline data. How are $a_1$ and $a_2$ subsequently updated or processed?**
>
> In lines 228-233, we introduce the training process of HM using a sub-goal $z_t$ as an example. Assume a c-step sequence $(s_g,s_t,a_t^*,s_{t+1},a_{t+1}^*,\dots,s_{t+c-1},a_{t+c-1}^*)$ exists in the offline data, where $s_g$ is the valuable sub-goal selected by the weighted average of accumulated rewards. In the training process, the Mamba model in HM generates a sub-goal $z_t$ and guides the transformer to predict the c-step action sequence (named $a_1$ here). Meanwhile, we replace the $z_t$ with $s_g$ and predict the c-step action sequence again (named $a_2$ here). Finally, we update the model parameters so that its predictions for these two action sequences ($a_1$ and $a_2$) are close to $(a_t^*,a_{t+1}^*,\dots,a_{t+c-1}^*)$. In the testing process, the trained HM is deployed in a new environment without parameter updates. This process will not have access to $s_g$, where HM relies on generating better sub-goals $z$ from the context to improve its performance.
>
> >**[7/7]W7.minor) The paper contains some typos and inconsistencies in tense usage. For example, in the related work section, the section on Mamba uses the present tense, while the section on in-context RL uses the past tense. These should be corrected for consistency. In addition, what's the meaning of the different gaussian distribution figures in Figure 1?**
>
> Thanks for the reviewer's corrections. We will revise the typos and inconsistencies in tense usage. In Figure 1 in the manuscript, the different Gaussian distributions indicate that Mamba generates a different subgoal for the transformer's predictions every c steps. It is possible to predict the sub-goal directly by using a fixed representation. However, sampling from a multi-variate Gaussian distribution introduces variability and flexibility in representing information. This approach can generate diverse sub-goals or latent variables, allowing for exploration and capturing more nuanced aspects of the input data.
>
> [1] In-context Reinforcement Learning with Algorithm Distillation. ICLR 2023.
> [2] Amago: Scalable in-context reinforcement learning for adaptive agents. ICLR 2024.

---

> > ### Author Response · Authors · 2024-08-12
> > **Looking forward to your reply**
> >
> > Dear Reviewer YUg4,
> >
> > We value your positive feedbacks and constructive suggestions for our paper and sincerely appreciate your effort in reviewing it. We hope we have effectively addressed all the concerns raised. As the end of the discussion is approaching, we are wondering if there are any additional potential clarifications or suggestions that you think would help us improve this manuscript.
> >
> > Thank you again for your dedicated review and invaluable insights.
> >
> > Kind regards,
> >
> > Paper5823 Authors

---

> > > ### Comment · Reviewer_YUg4 · 2024-08-13
> > > **Response to the authors**
> > >
> > > Thank you for your rebuttal. However, I have some reservations about the conclusion that the long-horizon experiments can validate HM's long-term memory capabilities. Additionally, considering the feedback from other reviewers, there are several incomplete or unclear parts in this paper. Therefore, I decided to maintain my original score at this stage.

---

### Official Review · Reviewer_wVw1 · 2024-07-09

**Soundness:** 4
**Presentation:** 4
**Contribution:** 3
**Rating:** 7
**Confidence:** 5

**Summary:**

This paper investigates to utilize the Mamba [1] architecture for In-Context RL task. Addressing this task with Transformer architecture is effective while it is very inefficient due to the quadratic computation overhead of Transformer. The Mamba can reduce this overhead dramatically while sustain the performance somewhat. The application of State-Space Models (SSMs) to In-Context RL task is studied in [2], but different from [2], they combinationally utilize Mamba and Transformer as high-level memory and low-level (short-term) memory. Additionally, as Mamba predicts the sub-goal for the Transformer short-term memory, they improved the performance. Through this modeling, they can achieve better performance than previous works while improving the efficiency.

[1] Gu, Albert, and Tri Dao. "Mamba: Linear-time sequence modeling with selective state spaces." arXiv preprint arXiv:2312.00752 (2023).

[2] Lu, Chris, et al. "Structured state space models for in-context reinforcement learning." Advances in Neural Information Processing Systems 36 (2024).

**Strengths:**

- Appropriate modeling is applied in this study. While the effectiveness of hybrid modeling of SSMs and local Attention has been previously explored in [1], the authors effectively implement this concept for the In-Context RL task with new functionalities, such as predicting high-value sub-goals.
- The introduction and methodology sections are well written. The motivation is clearly articulated, and the logical flow of their method proposal is coherent. The empirical analysis comparing Mamba and Transformer in RL tasks convincingly demonstrates the need for more advanced modeling.
- The paper provides extensive empirical analyses. It shares experimental results on multiple benchmarks, including ablation studies and performance changes with varying hyperparameter values.

[1] De, Soham, et al. "Griffin: Mixing gated linear recurrences with local attention for efficient language models." arXiv preprint arXiv:2402.19427 (2024).

**Weaknesses:**

- The high-level encoding is done by encoding the intervalled trajectories (e.g., every $c$ -th trajectory), which might miss important information in the middle of the interval.
- The section on Hybrid Mamba with Valuable Sub-goals is initially confusing, especially regarding the relationship between Mamba’s sub-goal prediction and the collected valuable sub-goals. Discussing this relationship at the beginning of the Valuable Sub-goal section could help readers understand the content more easily.
- One of the experimental results differs from my expectations, but the paper does not provide an analysis for this. I will address this in the Questions section.

**Questions:**

- I am curious why AD (transformer) shows worse performance than HM. I thought AD (transformer) performance would be the upper bound of HM while HM is more efficient. However, AD (transformer) performance is generally worse than HM in your tests, especially for Grid World in Figure 2. Why is AD (transformer) performance poor in Grid World? Did you use a smaller context size for the Grid World test? If not (using the same context size), what could be the reasons for the significant performance gap?

**Limitations:**

The authors properly addressed their limitations.

---

> ### Author Rebuttal · Authors · 2024-08-06
>
> We are particularly encouraged that Reviewer wVw1 finds our method effective.
>
> ### [I].Reply to the Weakness
> >**[1/2]W1. The high-level encoding is done by encoding the intervalled trajectories (e.g., every $c$-th trajectory), which might miss important information in the middle of the interval.**
>
> It is possible to provide complete trajectories to the high-level encoding. In fact, we tested non-intervalled trajectories during the paper's preparation and found that this setting did not produce the expected results. We design an HM variant that constructs a context sequence containing the complete trajectories but still generates subgoals for every $c$ steps.
>
> **Table 1. The performance of intervalled trajectories and non-intervalled trajectories.**
> | Trajectories | HalfCheetah-Med-Expert | HalfCheetah-Med| HalfCheetah-Med-Replay | Hopper-Med-Expert | Hopper-Med| Hopper-Med-Replay | Walker2d-Med-Expert | Walker2d-Med| Walker2d-Med-Replay |
> |-|-|-|-|-|-|-|-|-|-|
> | Intervalled trajectories | 96.12 $\pm$ 0.28 | 45.45 $\pm$ 0.35 | 45.26 $\pm$ 0.43 | 117.19 $\pm$ 0.65 | 83.15 $\pm$ 0.63 | 98.36 $\pm$ 0.51 | 118.21 $\pm$ 0.56 | 88.29 $\pm$ 0.76 | 95.66 $\pm$ 1.16 |
> | Non-intervalled trajectories | 95.98 $\pm$ 0.21 | 45.22 $\pm$ 0.18 | 44.97 $\pm$ 0.28 | 117.21 $\pm$ 0.61 | 82.98 $\pm$ 0.65 | 98.12 $\pm$ 0.38 | 117.97 $\pm$ 0.62 | 88.25 $\pm$ 0.68 | 95.62 $\pm$ 0.96 |
>
> Table 1 shows no significant performance gap between the two variants of HM. This is because (1) the transformer retains the complete short-term trajectory for action prediction. (2) The cross-episodic context in Mamba is long enough so that the uniformly sampled trajectory preserves sufficient information for predicting the sub-goals.
>
> >**[2/2]W2. The section on Hybrid Mamba with Valuable Sub-goals is initially confusing, especially regarding the relationship between Mamba’s sub-goal prediction and the collected valuable sub-goals. Discussing this relationship at the beginning of the Valuable Sub-goal section could help readers understand the content more easily.**
>
> Thanks for the reviewer's suggestion. We will explain their relationship in the revised manuscript. In summary, the valuable sub-goals collected can be regarded as additional signals, and we hope that the Mamba model can generate similar sub-goals to prompt the transformer's prediction.
>
>
> ### [II].Reply to the Questions
> >**[1/1]Q1. I am curious why AD (transformer) shows worse performance than HM. I thought AD (transformer) performance would be the upper bound of HM while HM is more efficient. However, AD (transformer) performance is generally worse than HM in your tests, especially for Grid World in Figure 2. Why is AD (transformer) performance poor in Grid World? Did you use a smaller context size for the Grid World test? If not (using the same context size), what could be the reasons for the significant performance gap?**
>
> As far as we know, AD was the first to introduce in-context learning into transformer-based RL. It uses the simplest (state, action, reward) tuples to form the context sequence, so it is not outstanding in performance and usually serves as a baseline for subsequent advanced methods, such as Amago [1] and DPT[2]. In the Grid World, both AD and our method HM set the same context that covers the same number of trajectories. This performance gap can be attributed to two aspects: token modality and model architecture.
> Our approach adds two different tokens compared to AD: the c-step cumulative reward and the done flag. Since the environment provided by Grid World is mostly sparsely rewarded, cumulative reward tokens are more advantageous. On the other hand,  recent methods, such as AT [3], demonstrated that done flag tokens are beneficial for in-context RL. In the model architecture, the hierarchical architecture of HM also brings about performance improvements, which is analogous to the fact that hierarchical RL learning is better at handling long-term and sparse reward tasks.
>
> According to the reviewer's suggestion, we will incorporate the above analysis into the revised manuscript.
>
>
> [1]Amago: Scalable in-context reinforcement learning for adaptive agents. ICLR 2024.
> [2]Supervised Pretraining Can Learn In-Context Reinforcement Learning. NeurIPS 2023.
> [3]Emergent Agentic Transformer from Chain of Hindsight Experience. ICML 2023.

---

> > ### Comment · Reviewer_wVw1 · 2024-08-11
> > **Reply to the rebuttal**
> >
> > Thank you the authors for your rebuttal. The authors properly addressed my concerns and I keep my score.

---

> > > ### Author Response · Authors · 2024-08-12
> > > **Appreciation to Reviewer wVw1**
> > >
> > > Dear Reviewer wVw1,
> > >
> > > We are grateful for your constructive suggestions and believe that incorporating the corresponding revision into the manuscript will significantly improve this paper. Thank you again for reviewing our paper and giving valuable feedback!
> > >
> > > Kind regards,
> > >
> > > Paper5823 Authors

---

### Official Review · Reviewer_yEub · 2024-07-10

**Soundness:** 2
**Presentation:** 2
**Contribution:** 2
**Rating:** 3
**Confidence:** 4

**Summary:**

The paper proposes Hybrid Mamba (HM) for in-context RL. Existing in-context RL methods are predominantly based on the Transformer architecture. Transformers come with quadratic complexity of self-attention and are computationally costly. Consequently, the authors propose a hybrid architecture that uses Mamba to compute sub-goals from long-context, which are fed into a low-level Transformer policy. The authors conduct experiments on grid-worlds and D4RL to evaluate their method.

**Strengths:**

**Relevance**

The paper aims at deploying the Mamba architecture for in-context RL, which is very relevant given the quadratic complexity of the Transformer architecture.
This results in clear benefits in terms of time complexity.

**Experimental results**

Empirical results on simple gridworld environments and D4RL seem convincing and their method exhibits significant gains compared to Transformers.

**Weaknesses:**

**Presentation**

The methodology raises some questions and should be improved, in particular:
 - What is the reasoning behind sampling the sub-goal from a multi-variate Gaussian?
 - How does this compare to using a fixed representation? (e.g., similar to CLS token)
 - Why is the done-flag in Hybrid Mamba necessary? Do other methods (e.g., AD [1]) use this as well?
 - What does “Extra high-value states” mean?
 - What is the intuition behind removing actions from the Mamba context?
 - What effect would dropping actions have in other methods?

Furthermore, the construction of “valuable sub-goals” is unclear.
One way to improve clarity would be to shorten the section on preliminaries and instead add more details to the Method section.
Figure 2 and Table 2 are missing the performance curves/scores for HM without valuable subgoals.
Finally, Figure 1 can be improved to enhance clarity.

**Significance of results**

The authors evaluate primarily on simple grid-world environments and rather simple robotics tasks. However, it is unclear how well HM generalizes to more complex tasks as used in other works [2].

**Evaluation**

The authors change their evaluation methodology from improvement curves on gridworlds (Figure 2) to average performance scores on D4RL (Table2).
On D4RL, HM seems to clearly outperform other methods.
However, the authors do not show in-context improvemenst which raises the question whether HM actually learns to improve in-context. Can the authors clarify, why no in-context improvement curves are shown for D4RL?

**Ablation studies**

Some ablation studies are missing and would add more depth to understanding the proposed method, in particular:
- What is the impact on performance of including the done-flag in Mamba?
- What effect does it have on other methods?
- What is the impact on performance of removing the action condition in HM?
- What effect does the same intervention have on other methods?


 [1] Laskin et al., In-context Reinforcement Learning with Algorithm Distillation, ICLR 2023
 [2] Raparthy et al., Generalization to New Sequential Decision Making Tasks with
In-Context Learning, ICML 2024

**Questions:**

- Did the authors consider techniques such as key-value caching for improving inference speed of Transformers for results reported in Table 4?
- Why is Mamba worse in effectiveness (Table 1)? What is a particular (theoretical) reason for this? Why does Mamba shorten the training time?
- How does performance generally change, when making the models bigger? Do bigger models help on these tasks? How large are the considered models?
- How well does the construction of valuable sub-goals generalize to other environments (e.g., with sparse rewards)?
- How do in-context improvement curves look like on D4RL?

**Limitations:**

The authors highlight that setting the context length c to a fixed value as a current limitation of their method. However, a notable limitations are missing, namely that their evaluation is limited to simple environments, while it is unclear how well HM performs on more complex or new tasks.

---

> ### Author Rebuttal · Authors · 2024-08-06
>
> Thanks for the reviewer's positive appraisal, insightful comment, and criticism of our paper.
>
> ### [I].Reply to the Weakness
> >**[1/6]W1.What is the reasoning behind sampling the sub-goal from a multi-variate Gaussian? How does this compare to using a fixed representation?**
>
> It is possible to predict the sub-goal directly by using a fixed representation. The multivariate Gaussian distribution is inspired by the method of predicting continuous variables in RL tasks, such as SAC[1]. In fact, sampling from a multivariate Gaussian distribution introduces variability and flexibility in representing information. This approach can generate diverse sub-goals or latent variables, allowing for exploration and capturing more nuanced aspects of the input data.
>
> >**[2/6]W2. Why is the done-flag in Hybrid Mamba necessary? Do other methods (e.g., AD) use this as well? What does “Extra high-value states” mean?**
>
> Our use of the "done-flag" is inspired by the Amago [2] and AT [3], which investigated the contexts consisting of different tokens and different numbers of episodes in D4RL. The ablation experiment found that the "done-flag" is beneficial for in-context RL to improve its performance from multiple historical trajectories.
>
> The "Extra high-value state" denotes the state with high-value weighted average of accumulated rewards in the future steps. During the training phase, we introduce extra high-value states to associate the actions generated by the transformer and assist the Mamba to generate similar sub-goals that align with these states. On the other hand, although the high-value sub-goal usually appears in the future time step of the current trajectory, it may also appear in the historical trajectory, encouraging the Mamba to extract information from the long sequence context.
>
> >**[3/6]W3. What is the intuition behind removing actions from the Mamba context?**
>
> We clarify that adding action tokens to the context of Mamba is possible. There are two reasons why we do not introduce action tokens: First, unlike transformers, the Mamba model does not generate actions, but rather representations of subgoals. Due to such subgoals do not appear explicitly in the training data, Mamba cannot use them as input for autoregressive generation. Therefore, we do not add action tokens to Mamba's input to distinguish this from the autoregressive generation of actions in the transformer. Second, our Mamba model predicts a sub-goal every $c$ steps, resulting in the context not being composed on consecutive time steps. Therefore, one-step actions can hardly help Mamba predict future states after multiple time steps, as they are used to assist in predicting the next adjacent state. In particular, we are not saying that actions are unimportant. In the context of consecutive time steps, actions are essential tokens, also included in our transformer and other baselines.
>
> >**[4/6]W4.The authors evaluate primarily on simple grid-world environments and rather simple robotics tasks. However, it is unclear how well HM generalizes to more complex tasks as used in other works.**
>
> The grid world is one of the most commonly used benchmarks for in-context RL, such as AMAGO[2], AD[4], and DPT[5]. This is because the grid world provides many tasks that are difficult to achieve zero-shot transfer, which is very suitable for testing the ability of methods to learn from context. In addition, we also tested on large variants of grid world, where the difficulty of the tasks is greatly increased. Figure 2 in our manuscript shows that the baselines all showed significant performance degradation in large variant tasks. We are also encouraged by the new work pointed out by the reviewers, published in ICML 2024, which exploits the potential of in-context RL in a completely different setting. Due to time constraints for the rebuttal, we will add their proposed new benchmark in the revised manuscript.
>
> >**[5/6]W5.On D4RL, HM seems to clearly outperform other methods. However, the authors do not show in-context improvemenst which raises the question whether HM actually learns to improve in-context.**
>
> In the current manuscript, we followed the prior in-context RL work by showing different styles of results for d4rl [3] and grid-world [4]. We supplement the D4RL plots in Figure 1 in the global pdf to avoid reviewer misunderstandings.
>
>
> >**[6/6]W6.What is the impact on performance of including the done-flag in Mamba? What is the impact on performance of removing the action condition in HM?**
>
> According to the reviewer's suggestions, we report the ablation studies on the done-flag and action condition. As we discussed in W2 and W3, the results in Table 1 show the corresponding conclusions. In addition, we show the improvement curve in Figure 1 in the global PDF.
>
> **Table 1. The ablation study on the done-flag and action condition.**
> |Token-flag|HalfCheetah-Med-Expert|HalfCheetah-Med|HalfCheetah-Med-Replay|Hopper-Med-Expert|Hopper-Med|Hopper-Med-Replay|Walker2d-Med-Expert|Walker2d-Med|Walker2d-Med-Replay|
> |-|-|-|-|-|-|-|-|-|-|
> | HM | 96.12 $\pm$ 0.28 | 45.45 $\pm$ 0.35 | 45.26 $\pm$ 0.43 | 117.19 $\pm$ 0.65 | 83.15 $\pm$ 0.63 | 98.36 $\pm$ 0.51 | 118.21 $\pm$ 0.56 | 88.29 $\pm$ 0.76 | 95.66 $\pm$ 1.16 |
> | HM without done-flag | 93.95 $\pm$ 0.26 | 42.21 $\pm$ 0.38 | 41.92 $\pm$ 0.36 | 112.21 $\pm$ 0.62 | 78.89 $\pm$ 0.58 | 93.21 $\pm$ 0.32 | 113.57 $\pm$ 0.75 | 86.24 $\pm$ 0.65 | 90.52 $\pm$ 0.66 |
> | HM with actions token | 96.03$\pm$ 0.12 | 45.51 $\pm$ 0.28 | 44.98 $\pm$ 0.18 | 117.22 $\pm$ 0.60 | 83.14 $\pm$ 0.65 | 98.25 $\pm$ 0.46 | 118.31 $\pm$ 0.58 | 88.15 $\pm$ 0.67 | 95.61 $\pm$ 1.05 |
>
> Due to word limitations, we will answer the remaining questions in the next comment.

---

> ### Author Response · Authors · 2024-08-06
> **Rebuttal of the remaining questions**
>
> Due to word limitations, we answer the remaining questions in this comment. Please review this part of the response after the rebuttal part.
>
>
> ### [II].Reply to the Questions
> >**[1/5]Q1. Did the authors consider techniques such as key-value caching for improving inference speed of Transformers for results reported in Table 4?**
>
> The transformer model we counted in Table 4 has applied key-value caching. When calculating the action of a new time step, we do not recalculate the key and value of the previous time step. However, the transformer's computational complexity grows quadratic with the sequence length, which makes it much slower than the Mamba method in the context of long sequences.
>
> >**[2/5]Q2. Why is Mamba worse in effectiveness (Table 1)? What is a particular (theoretical) reason for this? Why does Mamba shorten the training time?**
>
> Mamba models long sequence dependencies through an input-dependent selection mechanism. This mechanism acts independently on each token, so its computational complexity is less than the attention mechanism based on token pairs. Therefore, Mamba's computational complexity grows linearly with the sequence, making it naturally superior to transformers in terms of training speed and inference speed. However, even under the MDP assumption, the states in RL tasks are not independent, which causes Mamba to perform lower than the transformer in terms of effectiveness.
>
> >**[3/5]Q3. Do bigger models help on these tasks? How large are the considered models?**
>
> As shown in the hyperparameters in Table 3 of the manuscript, our Transformer model follows the setting of the AT method [3], and the Mamba model follows the setting of the DM method [6]. According to the conclusions in the Mamba paper, larger models generally improve performance. Due to resource and time constraints, we will add tests of larger models in the revised manuscript. However, it is worth mentioning that HM, with the current model size, can handle these tasks well.
>
> >**[4/5]Q4. How well does the construction of valuable sub-goals generalize to other environments (e.g., with sparse rewards)?**
>
> The construction of valuable sub-goals also generalizes well to environments with sparse rewards. Figures 2 and 3 in the manuscript report the tasks in a sparse rewards setting, showing sub-goals' effectiveness. In particular, we will try our best to test HM on the benchmarks suggested by the reviewers in the revised manuscript.
>
> >**[5/5]Q5. How do in-context improvement curves look like on D4RL?**
>
> As we clarify in W5, the curves of D4RL are shown in Figure 1 in the global PDF file.
>
> [1]Soft actor-critic: Off-policy maximum entropy deep reinforcement learning with a stochastic actor. ICML 2018.
> [2]Amago: Scalable in-context reinforcement learning for adaptive agents. ICLR 2024.
> [3]Emergent Agentic Transformer from Chain of Hindsight Experience. ICML 2023.
> [4]In-context Reinforcement Learning with Algorithm Distillation. ICLR 2023.
> [5]Supervised Pretraining Can Learn In-Context Reinforcement Learning. NeurIPS 2023.
> [6]Decision Mamba: Reinforcement learning via sequence modeling with selective state spaces. 2024.

---

> > ### Author Response · Authors · 2024-08-12
> > **Looking forward to your reply**
> >
> > Dear Reviewer yEub,
> >
> > We value your positive feedbacks and constructive suggestions for our paper and sincerely appreciate your effort in reviewing it. We hope we have effectively addressed all the concerns raised. As the end of the discussion is approaching, we are wondering if there are any additional potential clarifications or suggestions that you think would help us improve this manuscript.
> >
> > Thank you again for your dedicated review and invaluable insights.
> >
> > Kind regards,
> >
> > Paper5823 Authors

---

> > > ### Comment · Reviewer_yEub · 2024-08-13
> > >
> > > I would like to thank the authors for the detailed response and the additional results.
> > > My concerns about a more detailed presentation have been addressed, however some major issues still persist.
> > >
> > > **Significance of results**
> > >
> > > The empirical results are still limited to very simple RL tasks, other works in the field of in-context RL consider more complex tasks such as, e.g. Procgen [1], which also provide the possibility to evaluate on unseen seeds of the same task or entirely unseen tasks.
> > >
> > > [1] Raparthy et al., Generalization to New Sequential Decision Making Tasks with In-Context Learning, ICML 2024
> > >
> > > **Evaluation**
> > >
> > > One of my major concerns was regarding the lack of in-context evaluation. The whole premise of in-context learning is to learn **new** tasks only by providing them in the context of the model. The authors only show resuls for in-context improvement on the gridworld tasks as for D4RL they evaluate on the training task. The results on those tasks seem promising. However, since the man gist of the paper is in-context learning, I would recommend to add Procgen to their experiments and evaluate on unseen seeds and unseen tasks as in [1] to demonstrate how well HM is able to learn in-context. In order to progress the field, it is vital to move beyond mere gridworld based environments.
> > >
> > > Based on these reasons I have decided to keep my score.

---

### Author Rebuttal · Authors · 2024-08-06

Dear Reviewers,

We are very grateful to the reviewers for their valuable suggestions, which further improved our work. We provide the learning curve of our HM and ablation studies in d4rl tasks with a submitted 1-page pdf.

Thank you again for your careful review and helpful comments.

Kind regards,

Paper5823 Authors

---

### Decision · Program_Chairs · 2024-09-25

**Decision:**

Accept (poster)

**Comment:**

The authors propose a model called HM that combines transformers and the Mamba model to efficiently handle long memory in in-context reinforcement learning. HM generates high-value sub-goals through the Mamba model, which are then used to prompt the transformer for high-quality predictions.

The paper is well-written and easy to follow. The experiments are thorough and demonstrate strong empirical performance. While additional experiments on Procgen would be interesting, I believe the paper has already shown significant potential. Therefore, I recommend acceptance.